# PS1 Affects the Pathology of Alzheimer’s Disease by Regulating BACE1 Distribution in the ER and BACE1 Maturation in the Golgi Apparatus

**DOI:** 10.3390/ijms232416151

**Published:** 2022-12-18

**Authors:** Nuomin Li, Yunjie Qiu, Hao Wang, Juan Zhao, Hong Qing

**Affiliations:** 1School of Medical Technology, Beijing Institute of Technology, Beijing 100081, China; 2Key Laboratory of Molecular Medicine and Biotherapy, Department of Biology, School of Life Science, Beijing Institute of Technology, Beijing 100081, China

**Keywords:** BACE1, presenilin 1, mutants, endoplasmic reticulum, Golgi

## Abstract

Neuritic plaques are one of the major pathological hallmarks of Alzheimer’s disease. They are formed by the aggregation of extracellular amyloid-β protein (Aβ), which is derived from the sequential cleavage of amyloid-β precursor protein (APP) by β- and γ-secretase. BACE1 is the main β-secretase in the pathogenic process of Alzheimer’s disease, which is believed to be a rate-limiting step of Aβ production. Presenilin 1 (PS1) is the active center of the γ-secretase that participates in the APP hydrolysis process. Mutations in the PS1 gene (*PSEN1*) are the most common cause of early onset familial Alzheimer’s disease (FAD). The *PSEN1* mutations can alter the activity of γ-secretase on the cleavage of APP. Previous studies have shown that *PSEN1* mutations increase the expression and activity of BACE1 and that BACE1 expression and activity are elevated in the brains of *PSEN1* mutant knock-in mice, compared with wild-type mice, as well as in the cerebral cortex of FAD patients carrying *PSEN1* mutations, compared with sporadic AD patients and controls. Here, we used a *Psen1* knockout cell line and a PS1 inhibitor to show that PS1 affects the expression of BACE1 in vitro. Furthermore, we used sucrose gradient fractionation combined with western blotting to analyze the distribution of BACE1, combined with a time-lapse technique to show that PS1 upregulates the distribution and trafficking of BACE1 in the endoplasmic reticulum, Golgi, and endosomes. More importantly, we found that the *PSEN1* mutant S170F increases the distribution of BACE1 in the endoplasmic reticulum and changes the ratio of mature BACE1 in the trans-Golgi network. The effect of *PSEN1* mutations on BACE1 may contribute to determining the phenotype of early onset FAD.

## 1. Introduction

The β-amyloid precursor protein (APP) that accumulates in vulnerable brain regions in Alzheimer’s disease (AD) can be cleaved in two different pathways: amyloidogenic and nonamyloidogenic pathways, which include three different secretases: α-secretase, β-secretase, and γ-secretase. In the nonamyloidogenic pathway, APP can be cleaved sequentially by α- and γ-secretase. However, in amyloidogenic processing, APP can be cleaved sequentially by β- and γ-secretase to generate Aβ fragments, of which Aβ_38_, Aβ_40_, and Aβ_42_ are the three best known Aβ peptides [1]. β-Secretase BACE1 is a type I transmembrane aspartate protease containing 501 amino acid residues. The precursor mRNA can produce a variety of isomers of different enzyme activities by alternative splicing [2]. BACE1 is distributed in a variety of organelles, mainly in endosomes and the trans-Golgi network (TGN), because these two subcellular structures are in an acidic luminal environment, which is suitable for BACE1 to perform biological functions [3,4]. BACE1 subcellular localization is not simply due to the organelle’s pH, oligosaccharide modification, or the amount of APP substrate, it also influences BACE1′s location [1,4]. Studies have found that BACE1 protein matures through the secretory pathway [5,6]. It is initially synthesized as an immature protein precursor in the endoplasmic reticulum, and then the signal peptide and the N-terminal pro-peptide domain are cleaved and glycosylated by furin protease in the Golgi apparatus. It will eventually become an active enzyme, which can be transported to the plasma membrane and endosomes to play its biological role. The intracellular localization of BACE1 is regulated by a variety of factors, and the interaction between RTN protein and BACE1 can significantly increase the retention of BACE1 in the endoplasmic reticulum (ER), inhibit the cleavage of APP by BACE1, and reduce the production of Aβ [7].

γ-Secretase comprises four core subunits: presenilin 1 (PS1), nicastrin (NCT), anterior pharynx defective 1 (APH-1), and presenilin enhancer 2 (PEN-2), among which, PS1 is the catalytic component [8]. Mutations of the human PS1 gene (*PSEN1*) are a major cause of familial Alzheimer’s disease. Currently, more than 226 mutations have been identified in PS1 [9]. More *PSEN1* mutations will be reported over time. Remarkably, those early onset familial Alzheimer’s disease (FAD)-linked *PSEN1* mutations lead to increased molar ratios of Aβ_42_ over Aβ_40_ in cell lines and in the brains of transgenic animals, as well as in the plasma of FAD patients with *PSEN1* mutations, compared with controls [10,11,12]. Our previous study also confirmed that some mutations in *PSEN1* can alter the Aβ_42_/Aβ_40_ ratio in the 2EB2 cell line [13]. It has also been proposed that PS1 is implicated in regulating the intracellular trafficking and maturation of selected transmembrane proteins, such as βAPP, integrin beta 1, and nicastrin [14,15,16]. In addition, *PSEN1* mutations can increase the expression and activity of BACE1 [17] and compromise the maturation of BACE1 [18]. However, how PS1 affects BACE1 is not yet clear. PS1 is reported to be located in the plasma membrane, endocytic compartments, lysosomes, ER, and Golgi apparatus [19,20]. Previous results showed that PS1 and BACE1 can form a complex [21,22]. They colocalize in early endosomes and the Golgi apparatus [22].

Based on the above observations, we hypothesized that PS1 also influences the intracellular trafficking of BACE1. To clarify the effect of PS1 on BACE1, a wild-type (WT) and presenilin 1-deficient mouse embryonic fibroblast cell line (MEF) was used in the present study. Two mutations (D257A and S170F) in *PSEN1* were chosen to transfect the MEF cell line to prove the effect of PS1 on BACE1. Mutation at D257A is a negative control that does not lead to elevated secreted Aβ levels [23]. S170F in the PS1 gene is a well-known mutation causing AD at a very young age with rapid progression [24,25]. We chose this mutation to transfect the cell line, because it is one of the most aggressive forms of AD both clinically and pathologically. We showed that PS1 deficiency downregulates the expression of BACE1 and affects the distribution of BACE1 in different cell compartments. Taking the previous reports and our results together, we suggest that PS1 regulates the intracellular trafficking of BACE1.

## 2. Results

### 2.1. PS1 Overexpression Upregulated the Expression Levels of Endogenous BACE1 In Vitro

To investigate the effect of PS1 on the expression levels of BACE1, we transfected the PS1 plasmid (vector pcDNA4 as a control) into the MEF *Psen1* knockout cell line (PS1^−/−^). The PS1 protein levels are shown in Appendix A (western blots) and Appendix A (quantification analysis). The expression levels of BACE1 increased significantly (Figure 1A). Quantification of the western blot showed that overexpression of PS1 resulted in a 148% increase in BACE levels, compared with control conditions (*p* = 0.013, Figure 1B). The results of quantitative real-time polymerase chain reaction (qRT-PCR) were consistent with the western blot results (Figure 1C). In addition, the reduction in APP protein levels in the MEF PS1^−/−^ cells after overexpression of the PS1 wild-type plasmid was evidence of increased amyloidogenic processing.

To further confirm the regulation of PS1 on endogenous BACE1, the γ-secretase inhibitor L-685,458 was applied to block the activity of PS1. A total of 48 h after transfection with the PS1 plasmid, MEF PS1^−/−^ cells were treated with L-685,458 (final working concentration: 10 μM). Cells were collected for western blot analysis at 0, 60, 90, and 180 min after L-685,458 treatment (Figure 1D–H). Quantification of the western blot showed that when cells were treated with L-685,458, the BACE1 protein levels decreased over time, while the levels of APP increased over time (Figure 1E,F). The expression of BACE1 and APP was significantly altered in L-685,458-treated cells at 90 and 180 min. In addition, we detected the protein levels of the C-terminal fragments (CTFs) of APP hydrolysis cleavage to verify the inhibition ability of L-685,458. The results showed a significant increase in CTFs after the addition of the γ-secretase inhibitor, but no significant changes in PS1 protein levels when cells were treated with or without L-685,458 (Figure 1G,H). The results showed that the γ-secretase inhibitor L 685,458 could significantly inhibit the activity of γ-secretase, leading to an accumulation of CTFs in the cell. These findings supported the upregulation of BACE1 levels after PS1 overexpression in the MEF PS1^−/−^ cells.

### 2.2. PS1 Overexpression Upregulated BACE1 Colocalization with ER, Golgi/TGN, and Early Endosome

To determine how PS1 upregulates the expression levels of endogenous BACE1, many studies have been performed. There is a report stating that PS1 mutations can compromise the maturation of BACE1 [18]. Further studies have found that BACE1 protein matures through the secretory pathway [5,6]. It travels through the endoplasmic reticulum and the TGN, where it becomes an active enzyme and is eventually transported to the plasma membrane and endosomes to perform its functions. To investigate the effect of PS1 on the transportation of BACE1, we used sucrose density gradient centrifugation to separate organelles 48 h after transfection of the PS1 plasmid into the MEF PS1^−/−^ cell line. According to the subcellular location of BACE1, we chose three organelles to determine the distribution of BACE1 (the ER, endosome, and Golgi). The markers of the three organelles, i.e., Calnexin (marker of ER), EEA1 (marker of early endosome), Syntaxin-6 (marker of Golgi), and BACE1, were detected by WB. 

The results of sucrose density gradient centrifugation showed that when PS1 was transfected, BACE1 was enriched in the layer 3 and layer 4 fractions (Figure 2A,B). The layer 3 fraction was denoted as the ER, as indicated by calnexin. The layer 4 fraction was a mixture of the ER, early endosome, and Golgi. Although the ER and early endosome can be found in other layers, the Golgi was specifically enriched in layer 4. Studies have shown that BACE1 can form proBACE1 through a process of glycosylation in the ER, and then proBACE1 is transported to the TGN, where proBACE1 is catalyzed into mature BACE1 by Furin protease [26,27,28]. The mature BACE1 protein has stable properties and is then transported out of the Golgi/TGN and finally located in the cell plasma membrane and early endosome [18]. Figure 2C,D shows the BACE1 expression in the MEF PS1^−/−^ cell line according to the western blot of all fractions from sucrose density gradient centrifugation, showing the ratio of BACE1 expression in each layer to the BACE1 expression of whole layers. PS1 and BACE1 co-overexpression resulted in the retention of BACE1 in the ER, compared to BACE1 overexpression alone, suggesting that proBACE1 might prolong its dwelling time in the ER. The increase in the ratio of BACE1 location in the fourth layer fraction indicated that more mature BACE1 was produced from proBACE1 in the Golgi/TGN, which might eventually contribute to the higher BACE1 activity.

To confirm the colocalization between BACE1 and the cell apparatus, we used immunofluorescence staining to detect BACE1 and PS1 distribution in the MEF PS1^−/−^ cell line. The results showed that in MEF PS1^−/−^ cells, only a small ratio of BACE1 (green) protein colocalized with calnexin (red). When PS1 was transiently transfected into the MEF PS1^−/−^ cells, the colocalization ratio of BACE1 and calnexin significantly increased, indicating that BACE1 protein migrated to the ER (Figure 3).

Similar results were detected in the Golgi and early endosome (Figure 2). Overexpression of PS1 increases the colocalization of BACE1 and syntaxin-6 (red), as well as BACE1 and EEA1 (red). The Golgi is responsible for transporting, modifying, and packaging proteins synthesized in the endoplasmic reticulum and delivering the proteins to the targeted destinations. The results indicate that PS1 might benefit the processing, sorting, maturation, and transportation of BACE1 in the Golgi. 

Studies have demonstrated that the early endosome is the main site where BACE1 cleaves APP into CTFs [29,30]. Although we did not conclude from Figure 2 that PS1 affects BACE1 distribution in early endosomes, the data from immunofluorescence staining showed that BACE1 was rarely present in early endosomes without PS1 in cells (Figure 3), indicating that PS1 is essential for the colocalization of BACE1 and early endosomes. 

### 2.3. PS1 Overexpression Upregulated BACE1 Transportation to the ER and Golgi/TGN

To better analyze the transportation of BACE1 between the ER and Golgi, time-lapse imaging was applied to provide continuous filming for further analysis. We applied a Golgi tracker to stain the Golgi apparatus in living cells.

The half-life of BACE1 is approximately 8 h [31], so 8 h of continuous video was recorded to show the trafficking of BACE1. A total of 48 h after transfection with PS1, the colocalization of BACE1 and Golgi increased to 40%, which was 20% without PS1. A total of 3 h after time-lapse capture, that is, 51 h after transfection with PS1, the colocalization rate of BACE1 and the Golgi apparatus peaked at 48% (Figure 4). This result is consistent with the data of immunofluorescent staining. The colocalization of BACE1 and ER is similar to the colocalization of BACE1 and Golgi. A total of 48 h after transfection with PS1, the colocalization of BACE1 and ER increased to 14%, which was 3% without PS1 (Figure 5). At this time, cells already translated a large amount of BACE1; therefore, at 0 h of time-lapse capture, the colocalization of BACE1 and ER reached its peak. As time passed, BACE1 migrated to the Golgi and the early endosome from the ER, which is consistent with the results shown in Figure 2 and Figure 3.

Studies have revealed that BACE1 can be degraded in two ways: the ubiquitin-proteasome pathway and the lysosomal pathway [32]. Therefore, we detected the colocalization of BACE1 and the lamp (marker of lysosome, Appendix A)by time-lapse imaging to determine whether the distribution of BACE1 in the degradation pathways increases with PS1 transfection. However, the results did not show significant differences between cells treated with or without PS1. The colocalization of BACE1 and lysosomes was approximately 1.5% with or without PS1 (Appendix A).

### 2.4. PS1 Mutants Affect the Distribution and Maturation of BACE1 In Vitro

To further test whether FAD-related *PSEN1* mutations affect BACE1 trafficking from the ER to the Golgi, two FAD-related *PSEN1* mutation expression plasmids were constructed: S170F and D257A. S170F is a well-known mutation causing AD at a very young age with rapid progression, while D257A is often used as a negative control [24,25]. According to the reports, S170F can lead to increased Aβ peptide secretion and a reduction in the CTFs of APP, while D257A can lead to decreased Aβ peptide secretion and accumulation in the CTFs of APP [13,23]. Our western blot results showed that pPS1-S170F overexpression significantly increased the BACE1 protein levels (Appendix A), supporting these reports.

The western blot results of sucrose gradient fractionation showed that the subcellular location of BACE1 was altered by *PSEN1* mutations, compared to PS1 wild-type, as shown in Figure 2B. When the PS1 mutant D257A was transfected, the distribution of BACE1 increased markedly in the layer 5 fraction (Figure 6A). The WB results showed that the S170F mutant altered syntaxin-6 distribution toward the layer 3 fraction. With PS1 wild-type/D257A, or without PS1, syntaxin-6 was mainly enriched in the layer 4 fraction. The PS1 mutant S170F impeded the maturation of BACE1 in the layer 5 fraction (Figure 6B), compared to the negative mutant D257A and PS1 wild-type. The western blot of BACE1 showed two bands: the upper band is mature BACE1, and the lower band is immature BACE1. Compared to BACE1 expression in the MEF PS1^−/−^ cell line with BACE1 and D257A co-overexpression, the ratio of immature BACE1 expression in layer 5 increased dramatically with BACE1 and S170F co-overexpression. Taken together, the results indicate that PS1 plays a role in the Golgi and maturation of BACE1.

We also used immunofluorescence staining to detect the distribution of BACE1 after the transfection of *PSEN1* mutants in the MEF PS1^−/−^ cell line. The results showed that the colocalization ratio of BACE1 and calnexin was significantly increased in S170F-expressing cells, but decreased in D257A expressing cells (Figure 7), indicating that the *PSEN1* mutant S170F can increase the distribution of BACE1 in the ER. Similar results were observed in the Golgi (Figure 8), although the increase in BACE1 in the Golgi with *PSEN1* mutant S170F overexpression was not significant. Studies have reported that S170F can lead to increased Aβ peptide levels and a reduction in the CTFs levels [13,23]. These data provide an explanation for previous studies, showing that a greater distribution of BACE1 in the ER and Golgi might produce more Aβ peptide.

## 3. Discussion

In the pathogenesis of AD, β- and γ-secretases are two key secretases in amyloidogenic processing. Abundant research has been performed to reveal their roles in AD and their interaction mechanisms. It has been found that the Swedish mutation of APP (KM670/671NL) has a higher BACE1 affinity than the wild-type and can be cleaved by β-secretase during maturation (ER-TGN) [33], while most wild-type APP is cleaved by hydrolysis during endocytosis, especially in early endosomes [34]. First, the YENPTY domain of the APP cytoplasmic region and the Leu-Leu domain of the BACE1 cytoplasmic region play an important role in the endocytosis, sorting, and cleavage of APP. Second, APP interacts with BACE1 in endosomes. For example, both GGA1 (Golgi-localized γ-ear-containing ARF-binding protein 1) and GGA3 (Golgi-localized γ-ear-containing ARF-binding protein 3) play important roles in APP cleavage by BACE1. The downregulation of GGA3 not only increases the localization of BACE1 in early endosomes, but also inhibits degradation by inhibiting its transport to lysosomes, thus stabilizing the protein. These results indicate that the intracellular transport and subcellular localization of APP and BACE1 play an important role in the hydrolysis of APP. In this study, we demonstrated that PS1 influences the expression of BACE1 by sucrose density gradient centrifugation (Figure 2) and the trafficking of BACE1 by immunofluorescence staining (Figure 3). In particular, the time-lapse filming showed a vivid intracellular transportation trace of BACE1 in the ER and Golgi (Figure 4 and Figure 5).

The *PSEN1* mutation is known to be a key hereditary factor of FAD. Mutations in *PSEN1* can cause changes in γ-secretase activities. To date, increasing evidence has indicated multiple physiological roles of PS1, such as those in calcium homeostasis, neuronal development, neurite outgrowth, apoptosis, synaptic plasticity, and tumorigenesis [35]. In particular, the roles of PS1 in the pathogenesis of AD are a hotspot, such as those in Aβ production, the regulation of the Aβ_42_/Aβ_40_ ratio [36], and the effect on metabolism of APP [6]. However, little is known about its exact role of cellular transportation in AD. Furthermore, we checked FAD-related *PSEN1* mutations (D257A and S170F) to confirm the effect on BACE1 (Figure 6). All the results supported the suggestion that PS1 facilitates the trafficking of BACE1. 

A previous study proposed that *PSEN1* mutations can affect the expression, activity, and maturation of BACE1 [4,5,6], which is consistent with our results from sucrose density gradient centrifugation. Interestingly, the *PSEN1* mutation S170F attenuated the maturation of BACE1 (Figure 6B). The mechanism is unclear and needs further investigation. Several reports have shown that PS1 is associated with ER stress [37,38], while some did not observe ER stress fluctuation, with or without PS1 [39,40]. Furthermore, we confirmed that BACE1 was located mainly in the Golgi apparatus in MEF PS1^−/−^ cells (Figure 2). BACE1 has a low pH optimum of approximately 4.5, so it is predominantly located in acidic intracellular apparatuses, such as endosomes and Golgi [41]. According to our previous research, overexpression of the *PSEN1* mutation S170F can increase the C99 level and Aβ_42_/Aβ_40_ ratio and decrease the APP levels in a 2EB2 cell line that stably overexpressed Swedish APP and BACE1 [13]. Our data support the previous results. The major population of secreted Aβ peptides is generated within the TGN. APP can be transported in TGN-derived secretory vesicles to the cell surface. At the plasma membrane, APP is either cleaved to produce a soluble molecule, sAPP, or reinternalized within clathrin-coated vesicles to an endosomal/lysosomal degradation pathway. Vesical formation from the TGN and the ER was impaired in cells expressing FAD-linked *PSEN1* mutants, resulting in a reduction in APP delivery to the cell surface [14]. The APP cleavage pathway can occur in a variety of organelles, and β-Cleavage at the site between Met596 and Asp597 of APP (the Asp1 cleavage site) results in the release of C99 [1]. In addition to the Asp1 site, BACE1 can also cleave APP within the Aβ domain between Tyr606 and Glu607 (the Glu11 cleavage site). Preferential cleavage of APP by BACE1 at the Asp1 or Glu11 site is strongly dependent on the BACE1 subcellular localization and APP sequence close to the β-cleavage sites [1]. Accumulation of immature BACE1 in the ER can lead to β-cleavage of APP, predominantly at the Asp1 site, resulting in the release of C99. Compared with D257A, transiently transfected S170F can cause more BACE1 to stay in the ER to cleave APP at the Asp1 site. Here, we assume that *PSEN1* mutants increase BACE1 expression in the Golgi/TGN and provide a retention signal to stay in the Golgi/TGN, where PS1 and BACE1 cleave APP into Aβ peptides.

Much remains to be learned about the mechanisms by which PS1 regulates BACE1. More work is needed. Unfortunately, we did not obtain acceptable films of the trafficking effect of *PSEN1* mutants (S170F and D257A) on BACE1. Due to the poor condition of cells after PS1 mutant transfection, 4 h of continuous filming will cause a shift in focusing cells. Our present data preliminarily provide a novel alternative for the interaction of PS1 and BACE1. Secretase inhibitors targeting PS1 or BACE1, but that will not interfere with the other normal functions of both, are required to establish the therapeutic potential for AD.

## 4. Materials and Methods

### 4.1. Plasmid Construction

Wild-type PS1 human cDNAs (pPS1-WT) and pBACE1-N3-EGFP were obtained from Weihong Song lab (University of British Columbia, Vancouver, BC, Canada). Mutations in *PSEN1* (PS1-S170F and PS1-D257A) were generated by overlap extension PCR on the plasmid pcDNA4.1/PS1-WT using the following primers: 

PS1-WT: 5′-GGCGAATTCGTTCTAGATATAAAATTGATGGA-3′ and 5′-CCCAAGCTTAAATGACAGAGTTACCTGC-3′; 

PS1-S170F: 5′-GCAACAATAGAAATGATATAATAAGCCAGGC-3′ and 5′-GCCTGGCTTATTATATCATTTCTATTGTTGC-3′; 

PS1-D257A: 5′-CAGCCACTAAAGCATATACTGAAATCACAGCC-3′ and 5′-GGCTGTGATTTCAGTATATGCTTTAGTGGCTG-3′. 

### 4.2. Cell Culture and Transfection

The *Psen1* knockout embryonic fibroblast cell line (MEF PS1^−/−^) was obtained from the De Strooper laboratory (University College London, UK). *Psen1* knockout MEF cell lines were maintained in Dulbecco’s modified Eagle’s medium (DMEM, Gibco, Grand Island, NY, USA) containing 10% fetal bovine serum (FBS, Gibco, Grand Island, NY, USA), 100 IU/mL penicillin, 100 μg/mL streptomycin, pyruvic acid sodium salt (1 mM), glutamine (2 mM), 1% NEAA, 0.1 mM β-mercaptoethanol, and ESGRO recombinant mouse leukemia inhibitory factor (mLIF, 1000 U/mL, Sigma, St. Louis, MO, USA). For subcellular fractionation experiment and immunofluorescence experiments, Lipofectamine 2000 (Invitrogen Life Technologies, Carlsbad, CA, USA) was used for transient transfection of pcDNA4.1 or pPS1-WT constructs into MEF PS1^−/−^ cells, according to the manufacturer’s instructions. For the time-lapse experiment, pBACE1-N3-EGFP with or without PS1-WT was transiently transfected into the MEF cell line using electroporation. For the inhibition of γ-secretase activity, pPS1-WT was transfected into MEF PS1^−/−^ cells. A total of 48 h later, the γ-secretase inhibitor L-685,458 was added to the cell culture medium, reaching a final working concentration of 10 μM. MEF PS1^−/−^ cells were collected at 0, 60, 90, and 180 min after L-685,458 treatment.

### 4.3. Subcellular Fractionation

MEF PS1^−/−^ cells were grown on 100 mm diameter dishes at a density of 10 × 10^6^ cells per dish. Three dishes were transfected with PS1-WT, and three other dishes were transfected with pcDNA4.1. Sucrose gradient fractionation of raft-like proteins was conducted as previously described [42]. Cells were lysed using ice-cold cell homogenization buffer (0.25 M sucrose, 10 mM Tris, 1 mM EDTA, and 0.5 mM EGTA (pH 7.4)) and centrifuged at 1000× *g* for 2 min at 4 °C. The supernatant was collected by centrifugation at 8000× *g* for 20 min at 4 °C. The precipitate was resuspended in cell homogenization buffer and 100 mM sodium carbonate buffer and then sonicated in continuous mode for three 5 s bursts, with an output of 6 W. A total of 2 mL of 80% sucrose was added to each sample to obtain a final volume of 4 mL and a final concentration of 40% sucrose. The mixture was laid on 4 mL of 35% sucrose solution, followed by 4 mL of 5% sucrose solution, and then centrifuged at 180,000× *g* for 6 h using a swing out rotor (Beckman, Brea, CA, USA). After centrifugation, twelve 1 mL fractions were harvested from the top of the tube, and the fractions were subjected to western blot analysis.

### 4.4. Immunocytochemistry

MEF PS1^−/−^ cells were grown on confocal dishes at a density of 7 × 10^4^ cells per dish. After 24 h, the pcDNA4.1 plasmid, PS1 plasmid, or PS1 mutation plasmid was transfected. A total of 48 h later, the cells were fixed for 15 min in 4% paraformaldehyde and permeabilized with 0.2% Triton X-100 in PBS. Then, the cells were blocked with blocking solution (5% normal goat serum in PBST) for 1 h and incubated with primary antibody (BACE1, 1:500, R&D, Minneapolis, MN, USA; Calnexin, 1:200, Abcam, Cambridge, UK; Syntaxin-6, 1:200, Abcam, Cambridge, UK; EEA1, 1:200, Sigma-Aldrich, St. Louis, MO, USA; Lamp-1, 1:200, Abcam, Cambridge, UK) overnight at 4 °C in blocking solution. After washing with PBS, the cells were incubated with the fluorescent secondary antibodies: Alexa Fluor 488- or Alexa Fluor 647 for 1 h at room temperature in the dark. Hoechst33258 (ZSGB-BIO, Beijing, China) was used to stain the nucleus. Images were obtained by laser scanning confocal microscopy (Nikon, Tokyo, Japan). The colocalization data were quantified by FUJI software (Image J), which generated a Pearson’s correlation coefficient value.

### 4.5. Time-Lapse

MEF PS1^−/−^ cells were grown on confocal dishes at a density of 7 × 10^4^ cells per dish. After 24 h, PS1+ BACE1-N3-EGFP plasmids or pcDNA4.1+ BACE1-N3-EGFP plasmids were transfected. A total of 48 h later, the cells were washed twice with Hank’s solution containing calcium and magnesium (Solarbio, Beijing, China), and then tracker (ER-Tracker Red. 1:700, ZSGB-BIO, Beijing, China; Golgi-Tracker Red, 1:50, Beyotime, Shanghai, China; LysoTracker^®^ Red DND-99, 1:1000, Thermo Fisher Scientific, Waltham, MA, USA) was added to the dish. After incubation at 37 °C for 30 min, the working solution was discarded, and the cells were washed with Hank’s solution containing calcium and magnesium three times, followed by the addition of 10% fetal bovine serum-free medium without phenol red for observation under the microscope. A 100× objective lens with immersion oil was chosen for time-lapse imaging. The excitation light was 488 nm and 555 nm, the cell thickness was approximately 14 μm, and the Z-stack setting was 0.5 μm × 28 Z-slices in total. The duration of imaging was 8 h, with a 10 min interval of image acquisition. Imaris software was used to count the colocalization of BACE1 and different organelles in the cell.

### 4.6. Immunoblotting

Cells were lysed 48 h after transfection in RIPA Lysis Buffer (Applygen, Beijing, China) with 50 mM Tris-HCl (pH 7.4), 150 mM NaCl, 1% NP-40, 0.1% SDS, protease inhibitor cocktail (Roche, Base, Switzerland), and 4-(2-aminoethyl) benzenesulfonyl fluoride hydrochloride (AEBSF, Sigma, St. Louis, MO, USA). The samples were sonicated for 3 min and centrifuged at 14,000× *g* for 10 min at 4 °C, and the protein levels were determined by the Quick Start™ Bradford protein assay (Bio-Rad, Hercules, CA, USA). Cell lysates were subjected to SDS polyacrylamide gel electrophoresis (SDS-PAGE) with 16% Tris-tricine or 10% Tris-glycine gels. The samples were then transferred to PVDF membranes (Millipore, Burlington, CA, USA). For protein detection, membranes were blocked for 1 h with 5% skim milk and incubated with the primary antibody (BACE1, 1:1000, R&D, MN, USA; Calnexin, 1:3000, Abcam, Cambridge, UK; Syntaxin-6, 1:1000, Abcam, Cambridge, UK; EEA1, 1:1000, Sigma-Aldrich, MO, USA; Lamp-1, 1:1000, Abcam, Cambridge, UK; APP, 1:10,000, gift from Prof Weihong Song; PS1-NT, 1:500, Millipore, MA, USA; Actin, 1:6000, Sigma-Aldrich, MO, USA) at 4 °C overnight. After incubation with species appropriate secondary antibodies conjugated to horseradish peroxidase, immunocomplexes were visualized by chemiluminescence using an ECL (Thermo Fisher Scientific, Boston, MA, USA) system, and the intensities of the bands were quantified with Image Lab™ Software (Bio-Rad, Hercules, CA, USA).

### 4.7. Quantitative Real-Time Polymerase Chain Reaction (qRT-PCR)

qRT-PCR was used to analyze the mRNA levels of target genes using SYBR Green (Tiangen, Beijing, China). Total RNA was isolated by TRIzol (Invitrogen Life Technologies, Carlsbad, CA, USA) following the manufacturer’s instructions. cDNA was synthesized by FastKing -RT SuperMix (Tiangen, Beijing, China) following the manufacturer’s instructions. The results of three independent experiments were normalized, relative to that of GAPDH, to correct for differences in template input by applying the 2−ΔΔCt algorithm. The sequences of the primer pairs were as follows: *BACE1*: 5′-TGGAGGGCTTTGGAGGGCTTCTACGTTGTCTT-3′ and 5′-CATCATGGAAGGTTTCTATGTCGTCTTC-3′;

*GAPDH:* 5′-GACTTCAACAGCAACTCCCACTCTTCC-3′ and 5′-TGGGTGGTCCAGGGTTTCTTACTCCTT-3′.

### 4.8. Statistical Analysis

All statistical analyses were performed using GraphPad Prism 6 software (GraphPad Software, San Diego, CA. USA). Quantifications were performed from data generated during three independent experiments. Values represent the mean ± standard error. Comparisons of more than two groups were carried out using one-way ANOVA, and Dunnett’s post-hoc test was used to analyze PS1 construct values and the control group [43]. Statistical significance between two groups was determined by unpaired two-tailed t test. *p* < 0.05 was considered statistically significant.

## Figures and Tables

**Figure 1 ijms-23-16151-f001:**
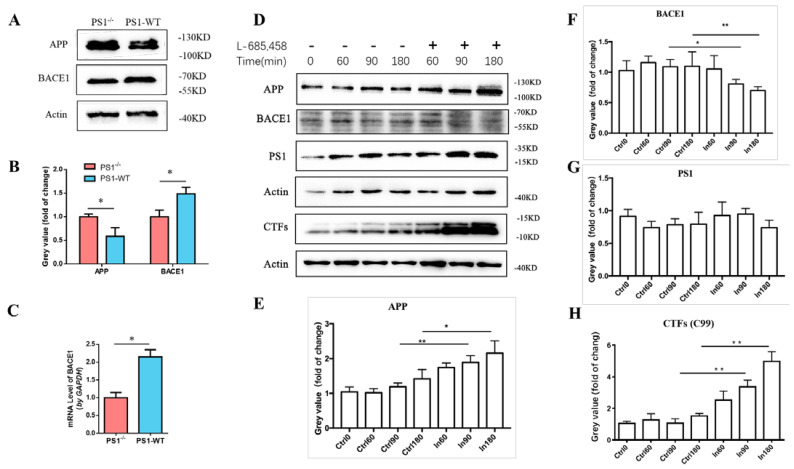
PS1 overexpression upregulated endogenous BACE1 in MEF PS1^−/−^ cells. (**A**) Western blot of the lysates 48 h after transfection with pPS1-WT or without transfection. The protein expression levels of APP and BACE1 were detected. (**B**) Quantification of APP and BACE1 in MEF PS1^−/−^ cells. (**C**) PS1 overexpression increased the mRNA level of BACE1 in MEF PS1^−/−^ cell. (**D**) Western blot of lysates from cells treated with γ-secretase inhibitor L-685,458 for 0, 60, 90, and 180 min after 48 h of transfection with pPS1-WT. The protein expression levels of APP, BACE1, PS1, and CTFs (C99) were detected. Quantification of APP (**E**), BACE1 (**F**), PS1 (**G**), and C99 (**H**); Actin was used as a reference; Dunnett’s post-hoc test: ** *p* < 0.01, * *p* < 0.05, (*n* = 3).

**Figure 2 ijms-23-16151-f002:**
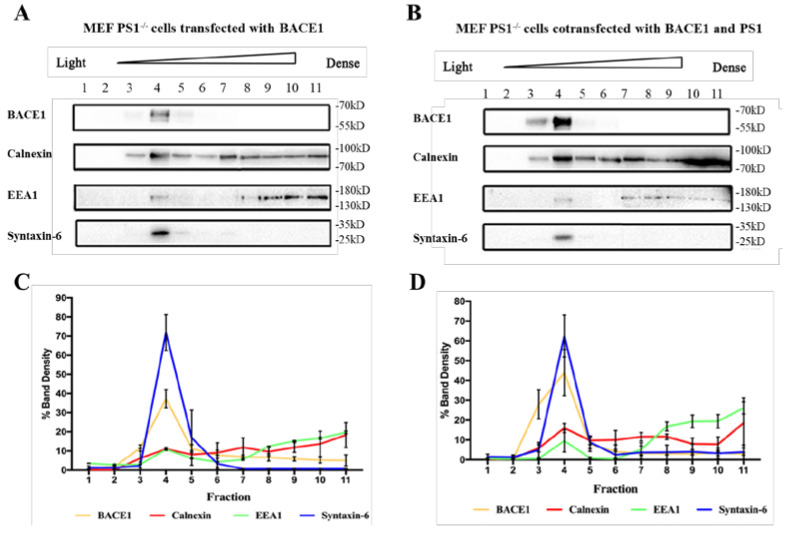
The effect of PS1 on BACE1 subcellular fractionation in MEF PS1^−/−^ cell. Western blotting of samples after sucrose density gradient centrifugation with transient transfection of BACE1 (**A**) or cotransfection of pBACE1 and pPS1 (**B**). Their quantitative analyses are shown in (**C**) and (**D**), respectively. The Y-axis represents the percentage of target protein in one fraction against total fractions (*n* = 3).

**Figure 3 ijms-23-16151-f003:**
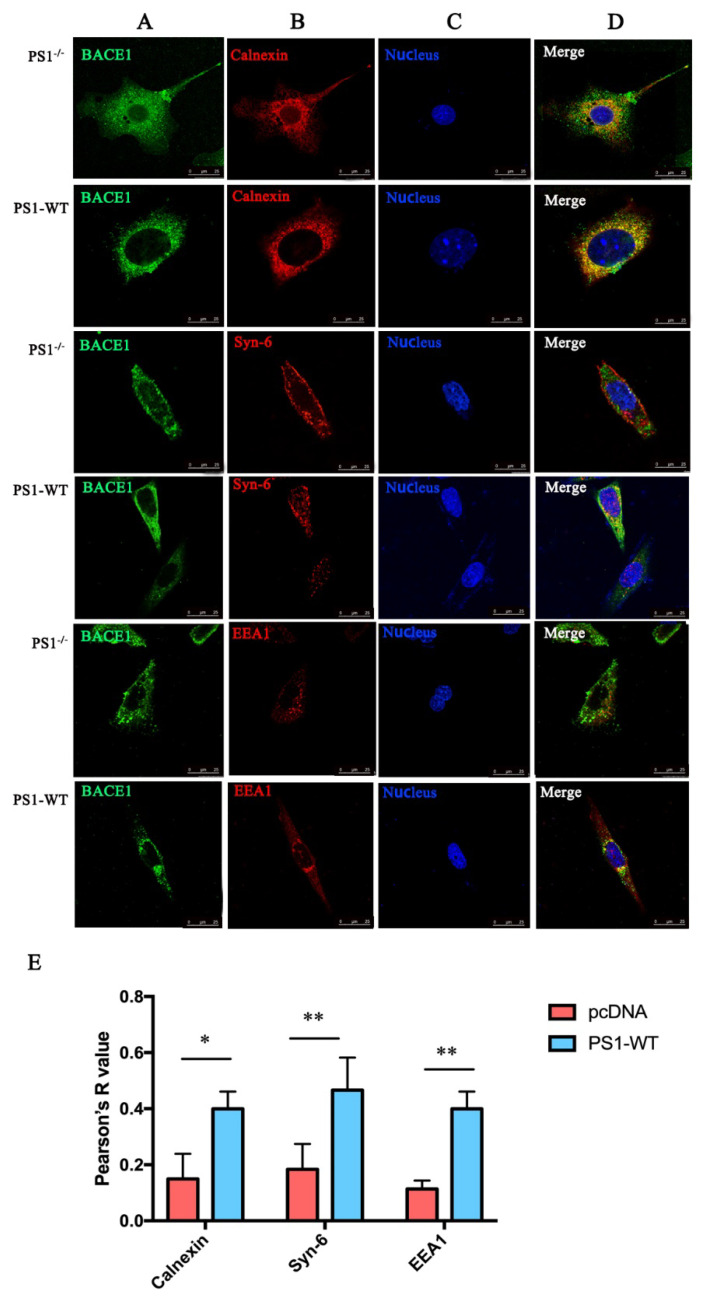
Intracellular distribution of BACE1 with or without PS1 (scale bar = 25 μm). (**A**) BACE1. (**B**) Organelle (calnexin: endoplasmic reticulum; Syn-6: Golgi; EEA1: early endosomes). (**C**) Nucleus (Hoechst33258). (**D**) Merged picture. (**E**) Quantification of the colocalization ratio. Dunnett’s post-hoc test: ** *p* < 0.01, * *p* < 0.05, (*n* = 3).

**Figure 4 ijms-23-16151-f004:**
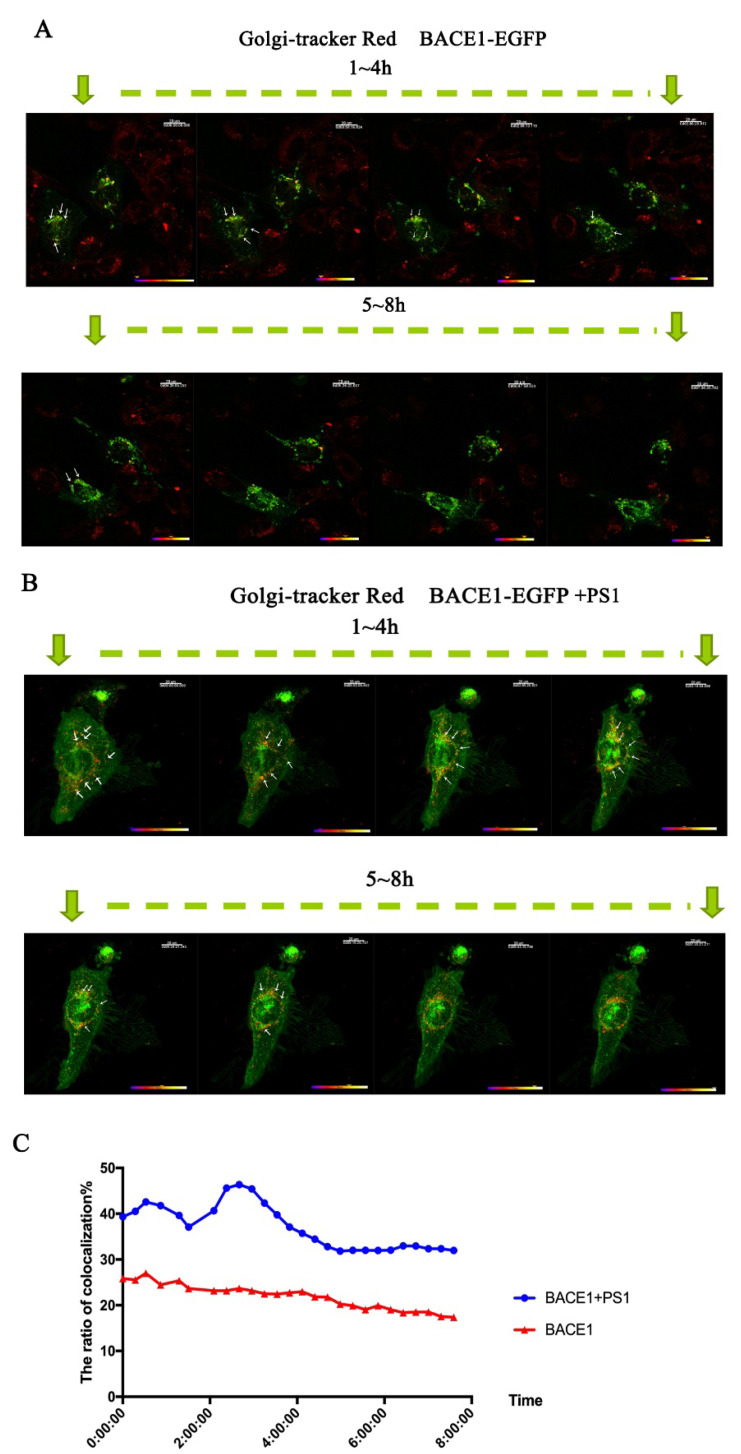
Time-lapse imaging of BACE1 and the Golgi tracker with or without PS1. (**A**) The distribution of BACE1-EGFP without PS1. (**B**) The distribution of BACE1-EGFP with PS1. (**C**) Quantitative analysis of the colocalization of BACE1-EGFP and the Golgi tracker with and without PS1.

**Figure 5 ijms-23-16151-f005:**
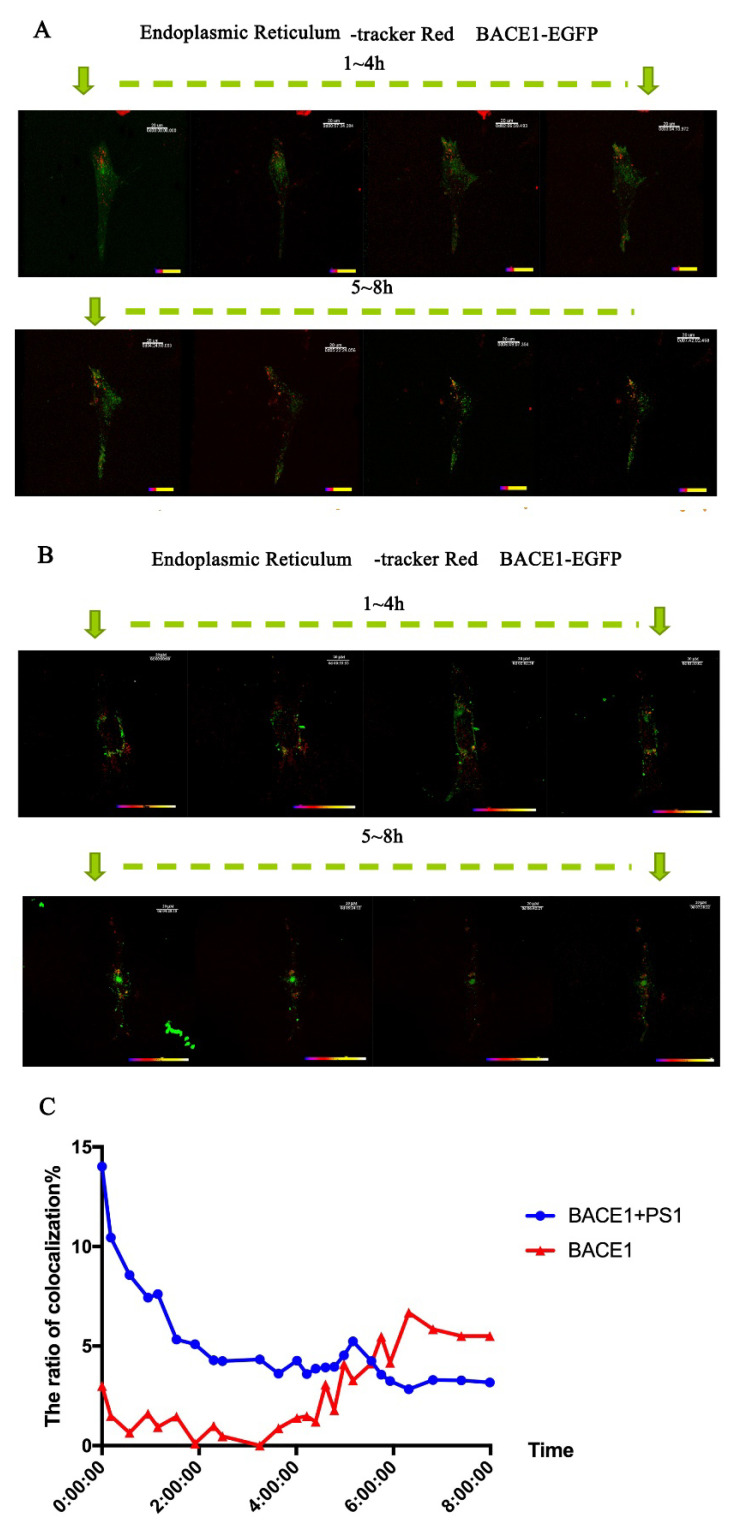
Time-lapse imaging of BACE1 and the ER tracker with or without PS1. (**A**) The distribution of BACE1-EGFP without PS1. (**B**) The distribution of BACE1-EGFP with PS1. (**C**) Quantitative analysis of the colocalization of BACE1-EGFP and the ER tracker with and without PS1.

**Figure 6 ijms-23-16151-f006:**
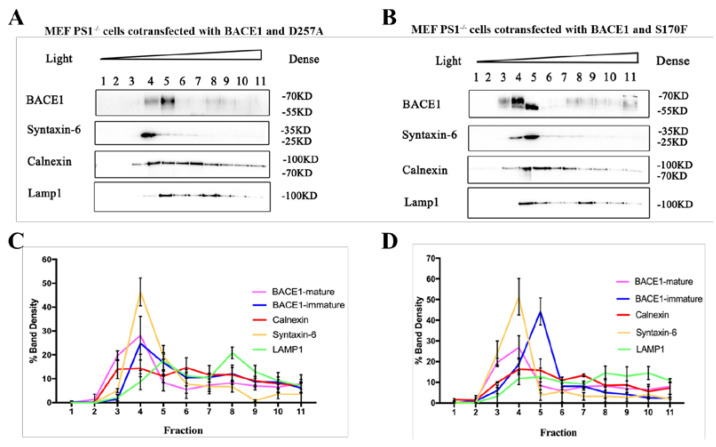
The effect of PS1 mutations on BACE1 subcellular fractionation in MEF PS1^−/−^ cell. Western blotting of samples after sucrose density gradient centrifugation with transient cotransfection of pBACE1 and pPS1-D257A (**A**) or pPS1-S170F (**B**). Their quantitative analyses are shown in (**C**) and (**D**), respectively. The Y-axis represents the percentage of target protein in one fraction against total fractions (*n* = 3).

**Figure 7 ijms-23-16151-f007:**
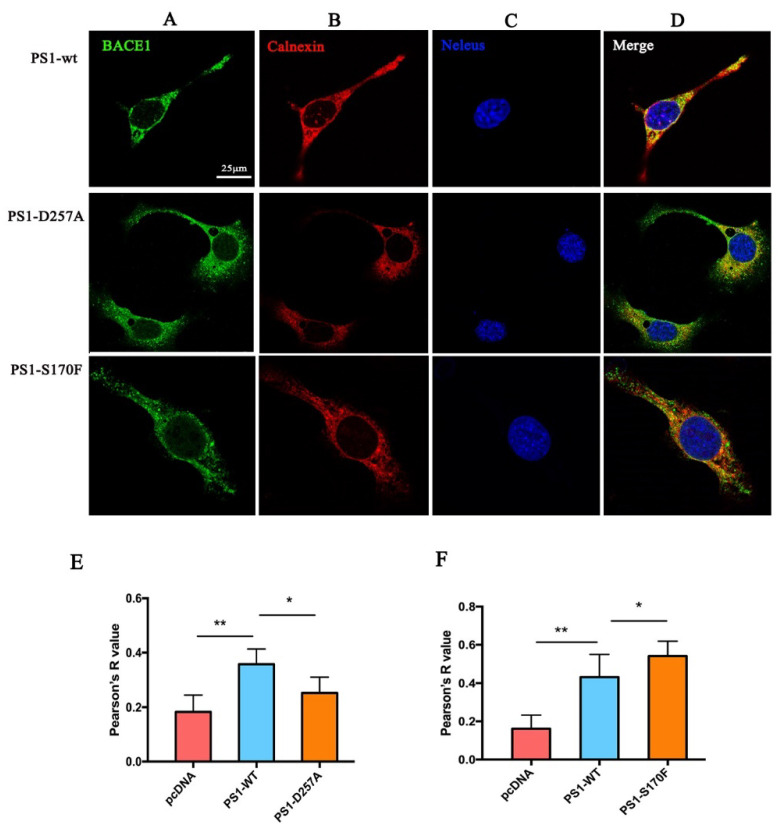
Effect of *PSEN1* mutations on colocation of BACE1 and the ER (scale bar = 25 μm). (**A**) BACE1. (**B**) Calnexin. (**C**) Nucleus (Hoechst33258). (**D**) Merged picture. Quantification of the colocalization ratio of BACE1 and the ER under *PSEN1* mutations D257A (**E**) and S170F (**F**). One-way ANOVA: ** *p* < 0.01, * *p* < 0.05 (*n* = 3).

**Figure 8 ijms-23-16151-f008:**
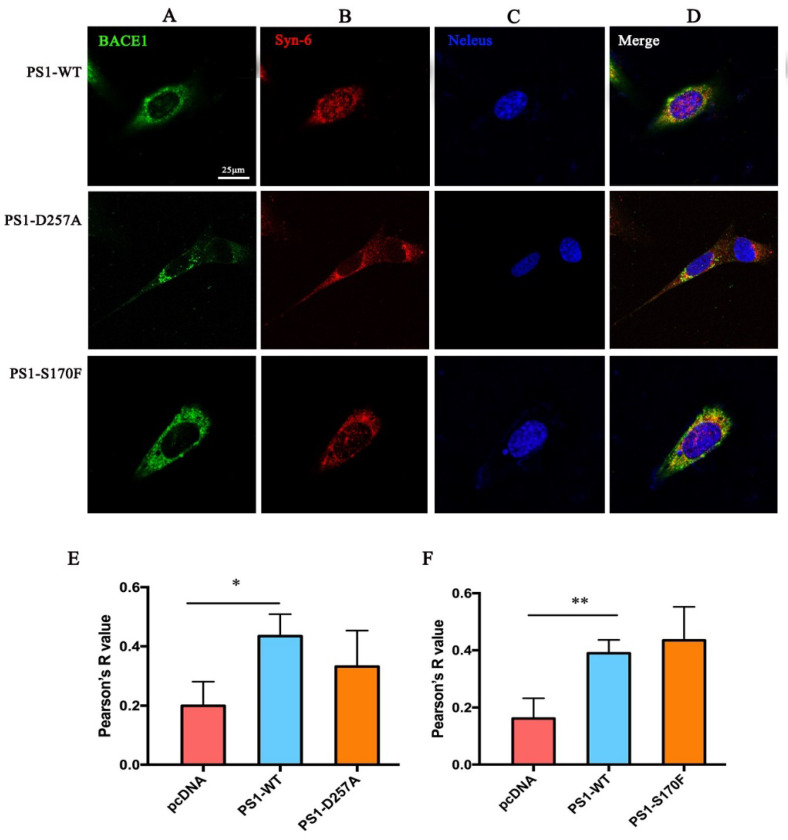
Effect of *PSEN1* mutations on colocation of BACE1 and the Golgi (scale bar = 25 μm). (**A**) BACE1. (**B**) Syn-6. (**C**) Nucleus (Hoechst33258). (**D**) Merged picture. Quantification of the colocalization ratio of BACE1 and the Golgi under the *PSEN1* mutations D257A (**E**) and S170F (**F**). One-way ANOVA: ** *p* < 0.01, * *p* < 0.05 (*n* = 3).

## Data Availability

The data that support the findings of this study are available from the corresponding author upon reasonable request.

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
