# Peer review of "PS1 Affects the Pathology of Alzheimer’s Disease by Regulating BACE1 Distribution in the ER and BACE1 Maturation in the Golgi Apparatus"

_ijms, 2022, doi:10.3390/ijms232416151_

Round 1

Reviewer 1 Report (Previous Reviewer 2)

The authors revised the manuscript adequately.

Author Response

Thank you.

Reviewer 2 Report (New Reviewer)

The study entitled „PS1 affects the pathology of Alzheimer’s disease by regulating 2

BACE1 distribution in the ER and BACE1 maturation in the 3

Golgi apparatus” by Li et al present a study, which shows that PS1 affects the expression of BACE1 in vitro in a knockout cell line. They also could show that PS1 upregulates the distribution and trafficking of BACE1 in the ER, Golgi and endosomes.

Major:

-I know the experiments are time consuming, but I feel n=3 is not sufficient to talk about “significant” results. Did those 3 experiments had multiple replicates in it? The authors should include this as a caveat of the study if so.

-The authors provide raw data of western blots – for figure one the authors have 4 couplets PS1/PS1-WT –are those included in the analysis of the data? Is it n = 4 then? If this is one n of each experiment, why do the authors only show one blot?

Minor:

-For better reading, the authors should rearrange the figures to each corresponding part in the results.

-The authors should add a little abbreviation of the denoted fraction to the corresponding number in Fig. 2 & 6. In addition, the used colors are hard to differentiate as a color vision deficient person (Syntaxin & BACE colors are too similar (at least for me)).

The concept and presentation of the study is well designed, but I feel like the sample size is the major weakness of the this study. This should be added as a caveat. In general this is in interesting study which will be a good read for researchers in the field of AD.

Author Response

Thank you for your valuable and suggestive comments.

Point-by-point addressing is listed below:

Manuscript ID: IJMS-2059792

The study entitled “PS1 affects the pathology of Alzheimer’s disease by regulating BACE1 distribution in the ER and BACE1 maturation in the Golgi apparatus” by Li et al present a study, which shows that PS1 affects the expression of BACE1 in vitro in a knockout cell line. They also could show that PS1 upregulates the distribution and trafficking of BACE1 in the ER, Golgi and endosomes.

Major:

-I know the experiments are time consuming, but I feel n=3 is not sufficient to talk about “significant” results. Did those 3 experiments had multiple replicates in it? The authors should include this as a caveat of the study if so.

Thank you. Usually, 3 independent experiments are fine for biological replicates. We did not include multiple replicates in 1 independent experiment this time. These replicates we call them technical replicates, they can help to reduce the variations. The data are consistent to previous, so we didn’t replicate the WB of the samples. For statistics, P < 0.05 was considered significant.

-The authors provide raw data of western blots – for figure one the authors have 4 couplets PS1/PS1-WT –are those included in the analysis of the data? Is it n = 4 then? If this is one n of each experiment, why do the authors only show one blot?

Thank you. Only the bands in the red box are the data for figure one. The raw data of other 2 Ns are as follows

Minor:

-For better reading, the authors should rearrange the figures to each corresponding part in the results.

Thank you. We use the IJMS template to prepare this manuscript. In the template, all figures are arranged together. But I guess the editor will rearrange the figures in publication version.

-The authors should add a little abbreviation of the denoted fraction to the corresponding number in Fig. 2 & 6. In addition, the used colors are hard to differentiate as a color vision deficient person (Syntaxin & BACE colors are too similar (at least for me)).

Thank you. We changed the colors, hope this time it is better for you. But we did not get the point “add a little abbreviation of the denoted fraction to the corresponding number in Fig.2 & 6.” Do you mean add the annotation in red to fig.2 & 6 as in the following picture?

Subcellular Fractionation of Hela Cells for Lysosome Enrichment Using a Continuous Percoll-Density Gradient. doi: 10.21769/BioProtoc.3362

Other pictures as references:

Regulated exocytosis of an H+ /myo-inositol symporter at synapses and growth cones. doi: 10.1038/sj.emboj.7600072.

Human NRAMP2/DMT1, which mediates iron transport across endosomal membranes, is localized to late endosomes and lysosomes in HEp-2 cells. doi: 10.1074/jbc.M001478200.

The concept and presentation of the study is well designed, but I feel like the sample size is the major weakness of the this study. This should be added as a caveat. In general this is in interesting study which will be a good read for researchers in the field of AD.

Thank you.

Round 2

Reviewer 2 Report (New Reviewer)

The authors adressed my comments. 

This manuscript is a resubmission of an earlier submission. The following is a list of the peer review reports and author responses from that submission.

Round 1

Reviewer 1 Report

In this manuscript, Li et al. reported that presenilin1 (PS1) overexpression induces expression (mRNA and protein) of BACE1 and also renders BACE1 more localized in ER and Golgi apparatus. Furthermore, exogenous expression of PS1 S170F that has a disease-causative mutation resulted in a larger increase in ER- and Golgi-localization of BACE1. Although the paper includes potentially interesting findings, the current manuscript contains a lot of issues to be clarified. Specific points are as follows.

Major points

1. The extent of upregulation of BACE1 protein should be compared between PS1 WT and a mutant. The authors showed that overexpression of PS1 WT increased BACE1 protein (mRNA as well) and altered BACE1 localization. While the effects of overexpression of PS1 mutant on BACE1 subcellular localization were examined, the level of BACE1 protein is not investigated.

2. Figure 1. As a critical issue, the expression level of exogenous PS1 is not provided. This must be shown. Figure 2 has also the same issue, and we do not know where PS1 is located and whether BACE1 and PS1 are co-localized.

3. Figure 6. It is interesting that a pattern of BACE1 bands in Figure 6B is quite different from the other samples. The authors described that the lower band in Fraction 5 is an immature form, but the identity of this form is not clear. I guess that this is possibly an ER glycoform. To test this possibility, the authors should compare the band patterns of BACE1 in the cell lysates with and without deletion of N-glycans by PNGaseF. 

4. Figure 2. According to the legend, the experiment was carried out three times, but no error bars are shown in the Figures. Without any error bar, we cannot see how reproducible the result presented in Fig. 2A,B. That is also the case for Figure 6.

5. Line 135-139. A sentence, “the lysosome and ubiquitin levels were detected” is not clear. What and how did the authors detect? I guess that “the lysosome” means co-localization analysis of BACE1 and LAMP1 by immunofluorescence. Although the authors described “not shown”, there is a related data in Fig. S1. Even worse, what is “BACE1 and ubiquitin change”? The authors should present the data or at least clearly describe what they did. One more thing, co-localization of a target protein with LAMP1 does not necessarily suggest an increased degradation of the target protein in lysosomes. To examine whether degradation of a target is enhanced or not, lysosome or proteasome inhibitors can be used which are both affordable.

6. Although the authors used Syntaxin 6 as a Gogi marker, Syntaxin 6 does not fully reside at the Golgi but rather is localized at both TGN and endosomes. The authors should confirm whether the similar result was obtained using another typical Golgi marker.

7. Line 128-134. The authors described “one possible reason” and “The other reason”, but these reasons for what? Please make the sentences clearer. If that was meant to be a reason for the increased expression of BACE1 protein, I wonder why the authors do not mention the upregulation of BACE1 mRNA shown in Fig. 1C. The authors should carefully interpret and integrate all the data to conclude how BACE1 protein expression is increased by PS1 overexpression.

8. Abstract. Compared with much background information, the results of the present study are little described. In particular, it is not described how the expression and trafficking of BACE1 are changed and regulated by PS1.

Minor points

9. Line 81-83 should be deleted.

10. Positions of molecular weight markers should be indicated in all the western blotting data. 

11. A space should be put just before the number of reference in the main text. That would make the paper more readable.

12. A period is lacking after “Fig” throughout the text.

13. A space should be added between value and unit.

14. Line 58. “Rations” should be corrected to “ratios”. Line 64, “affect” should be “affects”. Lin94, “upregulate” should be “upregulates”. Line 112, “Fruin” should be “Furin”. Line 140, “TNG” should be “TGN”, and many others. There are so many English editorial errors. I highly recommend that a native English speaker proofreads the paper before submission.

Reviewer 2 Report

In this study, the authors showed that PS1 regulates the expression and trafficking of BACE1 using wild-type and presenilin 1-deficient mouse embryonic fibroblast cell line (MEF). The effects of two PS1 mutations were also examined. The manuscript is well-written and the results are interesting. I have minor comments.

1. In Fig. 6A, the authors should indicate “MEF PS1-/- cells transfected with BACE1 and PS1-D257A”. Similarly, Fig. 6B should be “MEF PS1-/- cells transfected with BACE1 and PS1-S170F”.

2. Page 1, line 35, protein kinase => protease

Author Response

Point-by-point addressing is listed below:

Manuscript ID: IJMS-1873315

In this study, the authors showed that PS1 regulates the expression and trafficking of BACE1 using wild-type and presenilin 1-deficient mouse embryonic fibroblast cell line (MEF). The effects of two PS1 mutations were also examined. The manuscript is well-written and the results are interesting. I have minor comments.

  1. In Fig. 6A, the authors should indicate “MEF PS1-/- cells transfected with BACE1 and PS1-D257A”. Similarly, Fig. 6B should be “MEF PS1-/- cells transfected with BACE1 and PS1-S170F”.

Thank you. We changed Fig. 6. For details, please see Line 244.

  1. Page 1, line 35, protein kinase => protease

Thank you. “protein kinase” was changed to “protease” in Line 39.

Round 2

Reviewer 1 Report

The paper has been improved by the revision. However, there are several issues which have not been addressed. When the authors do not address any reviewer's request, the authors should logically explain a reason. The paper still requires substantial revision. Specific points are as follows.

1. In my previous comment 3, I asked the authors to examine whether the lower band is a glycoform or not using PNGase. However, any evidence or even logical speculation has not been provided in the response.

2. The authors added the error bars in Fig. 2 and 6, according to my previous comment. I was so surprised to see such small error bars in fractionation experiments, because I am sure that this kind of experiments is technically difficult. Are they independent experiments or just triplicates of the same solution? Please provide original images for 3 experiments.

3. It seems that the C5-ceramide data are missing, although the authors performed in this revision. Please provide it. 

4. It seems that Fig. 1D-H have been added in this revision. Please provide the reason for that.

Author Response

Response to Reviewer 1 Comments

Thank you for your valuable and suggestive comments.

Point-by-point addressing is listed below:

Manuscript ID: IJMS-1873315

The paper has been improved by the revision. However, there are several issues which have not been addressed. When the authors do not address any reviewer's request, the authors should logically explain a reason. The paper still requires substantial revision. Specific points are as follows.

  1. In my previous comment 3, I asked the authors to examine whether the lower band is a glycoform or not using PNGase. However, any evidence or even logical speculation has not been provided in the response.

Thank you. To address the first question: why not use glycosylation inhibitors for experiments? We have shown in the previous literature survey that BACE1 needs to undergo a series of complex post-translational modifications during the maturation process. Pro-bace1 is cleaved by Furin and other members of the Furin family to remove the 24-amino acids on N-terminal region of pro-peptide at the trans-Golgi network. Mature BACE1 has four N-glycosylation sites at Asn153, 172, 223 and 354, and BACE1 activity depends on the degree of N-glycosylation (10.1096/ FJ.05-5628com. 10.1074/ Jbc.m009361200, 10.1074 / JBC. M004175200.). In our study, we observed an upshift of BACE1 band in the fourth lane in the presence of PS1 mutation. At the same time, combined with the results of immunofluorescence, it was found that the localization of BACE1 in the Golgi apparatus was significantly increased under the condition of PS1 mutation, which is an important site for the posttranslational modification of BACE1. Therefore, we speculated that PS1 mutation promoted the glycosylation and other posttranslational modification of BACE1 and increased BACE1 activity and its ability to cleave APP at the Asp site. In this paper, we do not focus on which posttranslational modifications of BACE1 regulate BACE1 maturation. Therefore, no related glycosylation inhibitor treatment experiments were conducted.

  1. The authors added the error bars in Fig. 2 and 6, according to my previous comment. I was so surprised to see such small error bars in fractionation experiments, because I am sure that this kind of experiments is technically difficult. Are they independent experiments or just triplicates of the same solution? Please provide original images for 3 experiments.

Thank you. They are independent experiments. We got these data several years ago. We then checked the portable hard disk storing the data for the original images. Unfortunately, it is broken and the data in it is lost. We only have the raw pictures of Western Blot of one experiment and the excel containing grey analysis data for the whole 3 experiments of figure 2 and 6. So we cannot provide the original images for three experiments at present, but the data are true and reliable.

  1. It seems that the C5-ceramide data are missing, although the authors performed in this revision. Please provide it. 

Sorry, we didn’t make it clear. We didn’t add new experiment. When we did the time-lapse imaging to show the trafficking of BACE1 with or without PS1, we used the C5-ceramide, a Golgi tracker, to stain the Golgi in living cells (figure. 4). The colocalization of BACE1 and the Golgi increased markedly with PS1. It confirmed the results of immunofluorescence assay with Syntaxin-6 as the marker of Golgi.

  1. It seems that Fig. 1D-H have been added in this revision. Please provide the reason for that.

Thank you. The reason we added figure. 1D-H is to support that the PS1 regulates the BACE1 protein levels. We overexpressed PS1 in PS1 knock out cells, the BACE1 protein levels increased, then we inhibit the activity of overexpressed PS1, the BACE1 protein levels decreased. Figure 1D-H further confirmed the data of figure 1A-C.

Round 3

Reviewer 1 Report

Now most part of the paper looks fine except the point regarding Fig. 2 and 6. In the authors's response, they lost the original data and cannot provide them. In my opinion, this is not acceptable, but I cannot judge whether it is acceptable for this journal or not. Please follow editorial policy and decision.